# Psychometric Properties of the Authoritarian Attitude Scale in a Sample of Chilean Adolescent Students

**DOI:** 10.3390/bs15060756

**Published:** 2025-06-01

**Authors:** Ignacio Norambuena-Paredes, Karina Polanco-Levicán, Gustavo Troncoso-Tejada, Guillermo Davinson-Pacheco, Julio Tereucán-Angulo, José Luis Gálvez-Nieto, José Sepúlveda-Maldonado, Cristina Tavera-Cuellar, Adriana Bertoldi Carretto de Castro, Ítalo Trizano-Hermosilla

**Affiliations:** 1Departamento de Trabajo Social, Universidad de La Frontera, Temuco 4780000, Chile; ignacio.norambuena@ufrontera.cl (I.N.-P.); guillermo.davinson@ufrontera.cl (G.D.-P.); julio.tereucan@ufrontera.cl (J.T.-A.); 2Program de Doctorado en Ciencias Sociales, Universidad de La Frontera, Temuco 4780000, Chile; gustavo.troncoso@ufrontera.cl; 3Facultad de Educación, Universidad Autónoma de Chile, Temuco 4810101, Chile; karina.polanco@cloud.uautonoma.cl; 4Departamento de Educación, Universidad de La Frontera, Temuco 4780000, Chile; 5Departamento de Psicología, Universidad de La Frontera, Temuco 4780000, Chile; jose.sepulveda@ufrontera.cl (J.S.-M.); italo.trizano@ufrontera.cl (Í.T.-H.); 6Escuela de Administración de Empresas, MBA, Universidad Santo Tomás, Campus Bucaramanga, Bogotá 110110, Colombia; cristina.tavera@ustabuca.edu.co; 7Programa de Doctorado en Ciencias Económicas y Empresariales, Universidad de Granada, 18071 Granada, Spain; 8Faculdade de Tecnologia de Jahu (Fatec Jahu), Centro Estadual de Educação Tecnológica Paula Souza (CEETEPS), Jaú 17212-599, Brazil; adriana.castro@fatec.sp.gov.br

**Keywords:** authoritarianism, violence, racism, xenophobia, sexism

## Abstract

This study aims to evaluate the psychometric properties of the Authoritarian Attitude Scale in a sample of high school students from Chile. A total of 775 students (46.5% men; 53.5% women), with an average age of 15.99 years (Sd = 1.32), participated through non-probabilistic convenience sampling in 11 schools. Confirmatory factor analysis supported a three-factor structure with correlated factors and adequate fit indices. Criterion validity revealed positive and significant correlations with the violent behaviour dimension of the antisocial and delinquent behaviour scale. The factorial invariance analysis confirmed configural, weak, strong, and strict equivalence by gender, age and ethnicity, demonstrating model stability. The adoption of the Authoritarian Attitude Scale among Chilean adolescents provides a valid, culturally relevant tool for assessing authoritarian tendencies and understanding psychosocial dynamics in the educational context. The findings offer initial evidence of the scale’s reliability and validity.

## 1. Introduction

Authoritarianism is a central phenomenon in the social sciences, as it influences power, decision-making, and social relations ([27]; [73]). It has been studied from the perspective of social psychology ([9]; [29]), which analyses its cognitive and attitudinal foundations, and from the viewpoint of political science, which examines its expression in ideological regimes and movements ([6]; [11]; [64]). Beyond macrostructural aspects, the authoritarian attitude also has an impact on the individual level, shaping perceptions, values, and ways of interacting with others ([1]; [31]; [104]).

In adolescence, this attitude is associated with lower tolerance for diversity, cognitive rigidity, and blind obedience to authority ([41]; [85]). Its impact on socioemotional development influences conflict management, decision-making, and identity formation ([2]; [60]).

Authoritarianism is linked to a personality characterised by deep insecurity and lack of boundaries in structuring one’s own behaviours ([97]; [105]). The authoritarian attitude is associated with a set of beliefs that justifies inequality, violence, and social dominance, and its behavioural expression is manifested through discrimination, acts of violence against those perceived as weak or inferior, and the assertion of group superiority over those considered different ([3]; [19]; [29]). From this perspective, social dominance functions as a mechanism that reinforces existing hierarchies, legitimising coercive and exclusionary practices that perpetuate the subordination of certain groups based on criteria such as status, identity, or cultural belonging ([15]; [35]).

In a similar vein, various studies have shown that authoritarian attitudes are influenced by factors such as ethnicity, particularly in contexts of cultural diversity ([74]; [104]). This relationship becomes particularly relevant during adolescence—a key period in the formation of social and political identities ([20]). In Chile, and particularly in the Region of La Araucanía, where 34.3% of the population identifies with an Indigenous group, mainly Mapuche ([54]), it is pertinent to consider ethnic diversity ([17]; [107]).

### 1.1. Factors of Authoritarianism in Adolescence

Studies on authoritarian attitudes in secondary school students highlight authoritarianism as a key indicator for assessing beliefs and attitudes in adolescents ([55]; [106]; [116]). In this context, three main factors are addressed: sexism, justification of violence, and rejection of immigration ([21]; [83]). These dimensions not only conceptually align with the classic traits of authoritarianism, such as submission to authority, aggression toward outgroups, and adherence to rigid conventional norms ([1]; [4]) but also manifest in contemporary sociocultural contexts where adolescents construct their social identity ([68]). From this perspective, sexist beliefs, justification of violence, and rejection of immigration are valid indicators of authoritarianism and reflect an attitudinal framework shaped during adolescence, a stage highly influenced by social norms and relationships ([56]).

Therefore, the joint inclusion of these factors responds to a comprehensive understanding of authoritarianism as a socially shaped attitude, which is expressed in gender inequality, interpersonal violence, and ethnocentrism elements that have been widely recognised as problematic during adolescence and as robust indicators of authoritarian disposition ([21]).

Regarding sexist beliefs, this refers to the classification of men and women based on stereotypes and negative attitudes ([7]; [40]; [108]). This factor includes a hostile component, reflected in the perception of women as inferior ([67]; [95]). The concept of sexism is linked to discriminatory behaviours and attitudes based solely on a person’s sex ([36]; [49]; [70]). These prejudices are often expressed in subtle ways through symbols, language, and cultural customs ([16]; [82]). In this regard, they reinforce biases against women, which can lead to men’s reduced tolerance toward women, the perception that the advancement of women’s rights threatens men’s opportunities, and among women, the acceptance of submission, normalisation of violence, and limitation of their own freedom ([76]; [80]; [91]).

With regard to the justification of violence, it refers to the acceptance or legitimisation of intentional behaviours that cause harm or are detrimental to others ([71]; [72]; [117]). These behaviours can cause both physical and psychological harm, as they are based on the belief that violence is a legitimate means of resolving conflicts or asserting authority ([23]; [32]; [103]).

On the other hand, the rejection of immigration is based on exclusion, understood as marginalisation and social segregation ([37]; [101]). Immigration-related transformations are cross-cutting demographic processes that present various particularities and have a significant impact on the political, economic, cultural, and social spheres ([33]; [59]; [115]). It is important to consider that acceptance, rejection, tolerance, or intolerance towards immigration are influenced by the cultural capital and educational levels of the population engaging in discriminatory behaviour ([13]; [90]; [92]).

Studies on authoritarianism have shown a significant association between sexist beliefs and the justification of violence ([22]; [88]; [89]), with notable differences based on sex, men generally show higher levels of authoritarianism and sexist beliefs, linked to greater justification of violence, while women are more likely to reject it. These patterns reflect gender socialisation, which promotes dominance and aggression in men and empathy and co-operation in women. Structural factors like education and normative discourses also reinforce these differences ([10]; [57]; [84]; [99]).

In this regard, adolescence represents a crucial stage in psychosocial development, as it is the period when peer relationships are consolidated, and social norms and values are internalised, potentially influencing the reproduction or transformation of these beliefs ([14]; [102]; [118]). It has been observed that boys are more often associated with physical violence and its justification as a means of asserting social dominance ([109]), while girls show a stronger association with verbal violence and emotional manipulation ([42]; [48]; [111]).

Studies have indicated that sexist beliefs and the rejection of immigration share a common ideological basis linked to authoritarianism and the perception of cultural threat ([93]; [114]). Those who hold sexist beliefs often adhere to traditional views of social structure and gender hierarchy, which in turn may be associated with negative attitudes towards groups perceived as different or outsiders, such as immigrants ([26]; [28]; [75]). In this regard, the rejection of immigration can be explained, in part, by the resistance to changes in sociocultural values and norms that challenge the identity of the dominant group ([62]; [100]; [113]).

The justification of violence and the rejection of immigration may be inter-related through perceptions of threat and authoritarian attitudes ([30]; [53]). Various studies have indicated that individuals who justify the use of violence tend to display greater intolerance towards groups perceived as outsiders, including immigrants ([8]; [81]; [87]). This is due, in part, to the belief that immigration poses a risk to social, economic, or cultural order, which can lead to a greater acceptance of violent or punitive responses towards these groups ([39]; [77]; [94]).

Scientific literature has shown that the combination of sexist beliefs, justification of violence, and rejection of immigration among adolescents attending secondary education has multiple negative consequences ([58]; [63]; [69]). These attitudes contribute to a hostile school climate, where discrimination and exclusion affect coexistence and social cohesion ([44]; [50]; [52]). Furthermore, they reinforce gender stereotypes and xenophobia, which limits adolescents’ development in diverse environments and fosters the marginalisation of immigrant peers or vulnerable groups, and can even impact mental health ([51]; [61]).

The justification of violence is also linked to a higher risk of school bullying, especially towards students who challenge traditional gender roles or belong to minority groups ([46]; [66]; [96]). This not only impacts students’ emotional well-being, increasing levels of anxiety and stress, but also affects their academic performance, sense of belonging, and overall life satisfaction ([12]; [79]). At a collective level, the persistence of these attitudes hinders the integration of immigrant students and weakens schools’ ability to promote values of respect, empathy, and democratic coexistence, ultimately affecting social cohesion and young people’s preparedness for a diverse and inclusive society ([86]).

### 1.2. Measuring Authoritarianism in Adolescents

In this context, understanding and measuring these attitudes in adolescents is key to developing prevention strategies and promoting positive coexistence ([43]; [65]). For this reason, having validated instruments that can identify behavioural patterns associated with these beliefs is essential. One such example is the Authoritarian Attitudes Scale, developed with an adolescent sample in the Region of Murcia, Spain ([21]), and applied to a sample of 1960 secondary school students across 28 educational centres in the region.

Unlike other scales that measure the construct of authoritarianism, such as the Attitudes towards Diversity and Violence Questionnaire (CADV) by [34] ([34]), which contains 71 items and measures both beliefs and behaviours, the scale developed by [21] ([21]) is shorter and more specific. It efficiently targets beliefs and attitudes behind discriminatory and violent behaviours in adolescents. Unlike other scales, it integrates authoritarian dimensions into one tool, suited for educational assessment and prevention.

Therefore, the joint measurement of sexist beliefs, justification of violence, and rejection of immigration is especially relevant during adolescence, a stage in which social and political beliefs are consolidated and may persist into adulthood ([9]; [15]). These three factors represent contemporary expressions of authoritarianism that not only reflect intolerance and ideological rigidity but also contribute to the reproduction of dynamics of exclusion and discrimination both within and beyond the school setting ([19]; [69]). For this reason, having valid and reliable tools to identify these predispositions is essential for implementing preventive strategies aimed at promoting democratic coexistence, inclusion, and respect for diversity ([21]; [9]).

Although Spain and Chile share a language and some historical roots, cultural and linguistic differences affect instrument validation. In Chile, local idioms may influence item interpretation, and issues like gender gaps, school violence, and anti-immigrant attitudes persist among some youth ([111]; [52]).

Authoritarian attitudes characterised by obedience, normative rigidity and justification of force ([4]; [6]), have been linked to a greater acceptance of violence, particularly in threatening contexts. In adolescence, these attitudes often manifest in behaviours such as aggression, intimidation or weapon use, reflecting tendencies toward authoritarianism and social dominance ([2]; [5]). Violent behaviour and authoritarianism are distinct constructs, but they are related, as authoritarian attitudes can promote aggression toward those who challenge established norms or authority ([4]). In this sense, the link between authoritarianism and antisocial behaviour aligns with dual models of ideology and prejudice, which propose that the defence of rigid hierarchical structures is associated with greater permissiveness toward violence as a form of control ([15]; [35]).

This study aims to evaluate the psychometric properties of the Authoritarian Attitude Scale in a sample of Chilean high school students, examining its factorial structure, measurement invariance, criterion validity, and internal consistency. The use of this scale is crucial, given the need for culturally sensitive and psychometrically sound instruments to assess authoritarian tendencies during adolescence—a critical stage of psychosocial development. Such tools are essential for advancing sociopolitical psychology research and for informing educational and psychosocial interventions tailored to the local context. Additionally, the findings may guide the development of educational policies and programmes that foster a positive school climate and student well-being.

## 2. Materials and Methods

### 2.1. Participants

The study involved 775 secondary school adolescent students from public and government-subsidised schools in the Region of La Araucanía, Chile, corresponding to compulsory secondary education. To ensure data integrity, the QuestionPro platform was configured to require complete responses, thus, preventing the presence of missing values. This sample size allowed for sufficient variability for multivariate psychometric analysis and provided stability to the results.

The participants were selected through non-probabilistic convenience sampling. Eleven secondary schools that expressed their willingness to participate in the study were contacted, and students were invited to complete the online questionnaire during school hours, with prior authorisation from school authorities.

The general demographic’s characteristics are presented in Table 1. The sample consisted of secondary school students of both sexes (46.5% male and 53.5% female), aged between 14 and 19 years (M = 15.99, Sd = 1.32). These students were enrolled in eleven secondary education institutions in Chile.

### 2.2. Instruments

To achieve the study’s objectives, a sociodemographic questionnaire was administered to collect data on the students, including age, sex, educational level, and type of school, among other questions.

In addition, the Authoritarian Attitude Scale for adolescents was used (see Appendix A). This is a self-report instrument developed and validated in Spain ([21]). This instrument consists of 11 items rated on a 5-point ordinal scale (from 1 = strongly disagree to 5 = strongly agree). These items are grouped into three factors: sexist beliefs, justification of violence, and rejection of immigration. Reliability evidence ([21]), was measured as follows through Cronbach’s alpha coefficient: for the sexist beliefs factor, a Cronbach’s alpha of α = 0.75 was obtained, which is considered acceptable reliability; for the justification of violence factor, α = 0.70 also indicates acceptable reliability, falling at the minimum recommended threshold; and for the rejection of immigration factor, an alpha of α = 0.70 was reported, likewise reflecting acceptable reliability. These values indicate sufficient internal consistency for research purposes, although it is recommended that future studies consider improvements to enhance precision in certain factors. Regarding construct validity, the scale showed a three-factor structure that fit the data satisfactorily (RMSEA = 0.04; NFI = 0.98; CFI = 0.99).

The Antisocial and Delinquent Behaviour Scale was also administered. This is a self-report instrument developed and validated in a Spanish adolescent population ([5]). It consists of 25 items rated on a 5-point ordinal scale (from 1 = never to 5 = always). In the present study, the violent behaviour factor was used, which refers to engagement in criminal behaviours and the use of weapons. This factor is composed of six items, for example: “I have beaten up a stranger to the point of causing harm” or “I have attacked someone with a knife, stick, or another weapon”. Reliability indices for the violent behaviour factor in this study were acceptable: McDonald’s ω coefficient (0.847), Cronbach’s α coefficient (0.845), and GBL (0.875). The results of the confirmatory factor analysis indicated a good fit for the unidimensional model of the antisocial behaviour scale. The goodness-of-fit indices were satisfactory, with a GFI of 0.95, an AGFI of 0.94, and an RMR of 0.008, supporting the proposed structure and its factorial validity ([5]). These results suggest that the scale provides an adequate representation of the construct measured, reinforcing its usefulness in assessing antisocial behaviour in adolescents.

### 2.3. Procedure

First, five expert judges were selected to carry out a conceptual adaptation of the instrument based on the following criteria: specialised knowledge of the variable under study and lived experience in both cultural contexts (Chile and Spain). These criteria ensured an adequate understanding of the terms used in the items within the Chilean context. After the analysis, the judges concluded that no modifications to the instrument were necessary.

Prior to the administration of the instrument, approval was obtained from the Ethics Committee of the University of La Frontera (Project File Number UFRO No 156-19). Subsequently, school principals and parents or guardians were contacted, and their authorisation to carry out the study was obtained through the signing of informed consent forms.

Data collection was carried out through an online questionnaire hosted on the QuestionPro platform. Participants were invited to respond via emails sent by the research team, which included the informed consent previously signed by their parents or guardians, the informed assent, and the link to access the questionnaire. This document specifies the objectives of the study, the voluntary nature of participation, the confidentiality and anonymity of the data, the absence of risks, and the right to withdraw at any time. Data collection took place between September and December 2022.

### 2.4. Data Analysis

The first stage of the analysis was carried out using SPSS v.25 software and included the calculation of descriptive statistics, covering central tendency, dispersion, and shape, as well as the assessment of univariate and multivariate normality of the scale items. To provide evidence of validity, confirmatory factor analyses (CFA) were conducted using Mplus 7.11 software ([78]), employing a polychoric correlation matrix. In this process, the weighted least squares estimation method adjusted for mean and variance (WLSMV; [78]) was used.

Regarding fit indices, values equal to or greater than 0.90 on the comparative fit index (CFI) and the Tucker–Lewis index (TLI) were considered to reflect an adequate model fit ([98]). As for the root mean square error of approximation (RMSEA), values below 0.08 were taken as indicative of an acceptable fit ([18]).

Subsequently, factorial invariance analyses were also conducted using Mplus version 7.11, which included the following models ([112]): M0 configural (equal number of factors); M1 weak (equality of factor loadings); M2 strong (equality of thresholds); and M3 strict (equality of residuals). Factorial invariance was assessed following the recommendations of [24] ([24]), considering the following criteria of change: in CFI (ΔCFI) ≤ 0.01; in RMSEA (ΔRMSEA) ≤ 0.015; and in TLI (ΔTLI) ≤ 0.01 as evidence of invariance between the groups being compared.

Finally, the estimation of reliability was carried out using the JASP software version 0.19, applying different coefficients: McDonald’s omega (ω); Greatest Lower Bound (GLB); and Cronbach’s alpha (α), in accordance with the criteria established by [47] ([47]) and [110] ([110]).

## 3. Results

### 3.1. Descriptive Analysis

The descriptive analysis of the Authoritarian Attitude Scale applied to a sample of secondary school students was calculated based on the distribution of means observed in each item category. As shown in Table 2, item means ranged from 2.50 (Sd = 1.25) to 1.72 (Sd = 1.03); these values indicate that adolescents scored low on the scale. Additionally, the Kolmogorov–Smirnov (K-S) test was applied to assess the normality of the distribution for each item, yielding significant values (*p* < 0.01) in all cases. These results indicate that the distribution of the items deviates significantly from normality.

### 3.2. Factor Structure

A CFA was conducted with the eleven items of the instrument to evaluate the factorial structure of the Authoritarian Attitude Scale. The goodness-of-fit indices indicated an adequate model fit (WLSMV-χ^2^ [df = 41] = 115.477; *p* < 0.001; RMSEA = 0.048; 90% CI = 0.038–0.059; CFI = 0.986; TLI = 0.982), supporting the three-factor correlated solution: justification of violence; sexist beliefs; and rejection of immigration. The factor loadings showed adequate values on their respective factors, ranging from 0.592 to 0.924 (Figure 1).

### 3.3. Factorial Invariance

Once the structure of three correlated factors was confirmed, factorial invariance models were evaluated for the variables sex and age. As shown in Table 3, the scale reached a level of strict invariance for both variables. The first model tested was M0, which allows us to conclude that the factorial structure of the scale is the same for variables sex, age and ethnicity. Next, model M1 was tested, and it allowed us to conclude that the factor loadings are equivalent of the three evaluated variables. Then, model M2 was tested, which imposed constraints on item thresholds; the results indicate that the thresholds are equivalent across the three variables. Finally, model M3 was tested, which constrained the residuals; the results indicate that item residuals have equivalent magnitudes across sex, age and ethnicity.

### 3.4. Criterion Validity

Additionally, bivariate correlations were evaluated between the violent behaviour factor of the antisocial and delinquent behaviour scale and the three factors of the authoritarian behaviour scale in adolescents. The results showed that the violent behaviour (mean = 7.25; Sd = 2.74) factor correlated positively and significantly with justification of violence (r = 0.315; *p* < 0.001; mean = 9.67; Sd = 2.59), sexist beliefs (r = 0.217; *p* < 0.001; mean = 7.28; Sd = 3.47), and, finally, rejection of immigration (r = 0.197; *p* < 0.001; mean = 6.25; Sd = 2.91). Although the correlation coefficients show small to moderate effect sizes according to conventional criteria ([25]), these results are consistent with expectations in psychological and social research, where constructs are often complex and influenced by multiple factors ([45]). It is important to highlight that the statistical significance and theoretical coherence of these associations support the interpretation that individuals with higher scores in violent behaviour also tend to exhibit higher levels of attitudes that legitimise violence and reflect authoritarian, or exclusionary views of society. The effect sizes are relatively small, this is expected given the complexity of the constructs involved. Even modest associations can yield meaningful insights when they are theoretically grounded and statistically significant. Although the effect sizes are modest, their statistical significance and conceptual alignment support the presence of consistent patterns in the variability between the evaluated factors ([21]; [45]).

### 3.5. Reliability Evidence

Next, the reliability evidence of the Authoritarian Attitude Scale was analysed in a sample of secondary school students. As shown in Table 4, the three factors showed satisfactory results, with the rejection of immigration factor standing out as the most reliable according to McDonald’s omega coefficient.

## 4. Discussion

The objective of this study was to evaluate the psychometric properties of the Authoritarian Attitude Scale in a sample of high school students from Chile. The findings of this study provide robust empirical evidence regarding the validity of the Authoritarian Attitude Scale in this population, confirming its factorial structure composed of sexist beliefs, justification of violence, and rejection of immigration. This factorial configuration is consistent with the original proposal by [21] ([21]), they developed and applied the scale in an adolescent population, demonstrating its conceptual stability and applicability across different contexts. Despite the shared language between Spain and Chile, a linguistic test was conducted to account for contextual variations. Confirmatory factor analysis supported the three-factor correlated model, with satisfactory fit indices, validating the proposed structure.

Confirmatory factor analysis supported the three-factor correlated model, validating the structure of the Authoritarian Attitudes Scale. The dimensions justification of violence, sexist beliefs, and rejection of immigration showed strong loadings and internal consistency, reinforcing the instrument’s structural validity ([21]).

Beyond statistical confirmation, these results allow for an integrated understanding of how these three factors form an authoritarian attitudinal core, characterised by the legitimization of violence as a mechanism of control, the reproduction of gender hierarchies, and the rejection of cultural otherness ([91]; [19]). The observed interrelation among the factors suggests that these dispositions do not operate in isolation but are part of an ideological constellation that may influence how individuals interpret and position themselves in relation to diversity and authority ([15]; [29]).

From a broader perspective, the empirical evidence obtained provides a solid theoretical and methodological framework for studying authoritarianism in adolescents, a critical stage in the consolidation of beliefs and values ([21]; [55]). Furthermore, it strengthens the instrument’s potential to be used in educational contexts for diagnostic, formative, and preventive purposes, as well as in cross-cultural studies aimed at analysing contemporary expressions of authoritarianism and their implications for democratic coexistence and respect for human rights ([9]; [29]).

The factorial invariance analysis by sex, age, and ethnicity confirmed equivalence across all parameters, including factor structure, loadings, thresholds, and residuals. Achieving strict invariance ensures that score differences reflect true variations in authoritarian attitudes, not measurement bias. This advances the original scale by validating its comparability across groups.

Regarding criterion validity, the factors of the Authoritarian Attitude Scale showed significant correlations with the violent behaviour factor, especially justification of violence, followed by sexist beliefs and rejection of immigration. These results align with prior research linking authoritarian attitudes to increased tolerance and normalisation of violence in adolescents ([5]).

These findings support previous research showing that authoritarian beliefs are linked to the acceptance of violence as a legitimate form of control, reinforcing dominance–submission dynamics and fostering both overt and subtle forms of aggression and inequality among adolescents ([5]; [38]; [91]).

On the other hand, the reliability of the scale was adequate for each of the evaluated factors, with McDonald’s omega coefficients above 0.70. In particular, the rejection of the immigration factor showed the highest level of internal consistency (ω = 0.801), indicating substantial homogeneity among the items that comprise it. These results support the psychometric stability of the scale and its suitability for use in future studies aiming to analyse authoritarian attitudes in adolescent populations for comparative or longitudinal purposes ([2]; [55]).

This study provides strong empirical evidence on the factorial structure, invariance, criterion validity, and reliability of the Authoritarian Attitude Scale among secondary students, confirming its psychometric robustness. The findings enable more precise analysis of how authoritarian attitudes relate to psychosocial factors like radicalisation, violence, and socialisation in diverse contexts ([30]).

### Limitations and Directions for Future Research

Another important limitation of this study is the limited use the scale has had in different contexts, which restricts the availability of broader empirical evidence regarding its validity and reliability across diverse populations. Although the scale has demonstrated adequate psychometric properties in the samples analysed ([21]), its applicability in other sociocultural settings still requires further exploration.

A key limitation of this study is the inability to test alternative models or allow residual correlations between items, potentially affecting model fit precision. Future research should increase the number of items per dimension and explore alternative structures such as unidimensional, hierarchical, or bifactor models to reinforce structural validity. It is also recommended to conduct follow-up or test–retest studies to assess temporal reliability, particularly given the developmental changes that may influence adolescent attitudes over time.

One limitation of the instrument is the absence of a specific dimension addressing racial bias, which may constrain its conceptual validity in culturally diverse contexts.

An important direction for future research is the replication of the scale in other Latin American countries to evaluate its stability across diverse sociocultural contexts. Cross-national invariance studies would help determine whether the scale measures the same constructs consistently across different educational and cultural settings ([63]; [97]).

The inclusion of delinquent behaviours as a criterion for criterion validity is grounded in the literature recognising its relationship with authoritarian dispositions, particularly in their punitive and social control dimensions ([2]; [5]). Its use made it possible to explore a behavioural manifestation of the construct within highly vulnerable school contexts.

Likewise, the cross-cultural validation of the scale would not only strengthen its psychometric robustness but also contribute to the development of standardised instruments for measuring authoritarian attitudes in secondary school students within the region. This would facilitate international comparisons and broader research on the factors that influence the adoption of authoritarian attitudes in different sociopolitical contexts ([19]; [27]; [63]). Ultimately, expanding the empirical base of the scale would enable its use in intervention studies and educational policies aimed at promoting democratic values and preventing the development of authoritarian attitudes among youth ([13]; [69]; [86]).

Likewise, future research could develop more comprehensive models that incorporate these variables, allowing for a deeper and more detailed analysis of the factors influencing authoritarian attitudes in secondary school students ([21]). The inclusion of additional variables, such as socioeconomic factors, family influences, educational experiences, and exposure to political or ideological discourse, would enable a more precise understanding of the underlying determinants of this construct ([13]; [83]; [105]).

Based on these findings, the study supports the notion that the dimensions of authoritarianism (sexism, justification of violence, and rejection of immigration) form an interconnected belief system rather than independent manifestations. Their convergent association with violent behaviour suggests the presence of a coherent attitudinal structure that underlies various expressions of prejudice and social exclusion ([21]).

## 5. Conclusions

The findings of this study confirm the validity and reliability of the Authoritarian Attitude Scale in secondary school students in Chile, supporting its psychometric robustness and its usefulness in research on school coexistence and authoritarian attitudes. Moreover, the results provide evidence of configural, weak, strong, and strict invariance of the model, supporting different levels of equivalence between the groups analysed.

The confirmation of strict factorial invariance across sex, age, and ethnicity indicates that the scale measures the construct equivalently across groups. This progression from configural to strict invariance ensures comparability in factor structure, loadings, intercepts, and residuals, allowing for unbiased comparisons of latent means and structural relationships in research on authoritarian attitudes among adolescents from diverse sociodemographic backgrounds.

Future research should assess the stability and generalizability of these findings in larger, more diverse samples and explore model invariance in complex analytical structures. This would support the development of precise instruments suited to Latin American educational contexts and enable their use in comparative and longitudinal studies to track changes over time.

In this regard, the present study provides essential empirical evidence for a deeper understanding of authoritarian attitudes in secondary school students, contributing to the strengthening of the scale as a valid and reliable measurement instrument for analysing this issue and its impact on school coexistence dynamics. The application of this scale in future research will allow not only for a more accurate assessment of the factors associated with such attitudes but also for the identification of underlying patterns that could inform the design of pedagogical interventions and educational policies aimed at promoting more democratic, inclusive, and diversity-respecting school environments.

## Figures and Tables

**Figure 1 behavsci-15-00756-f001:**
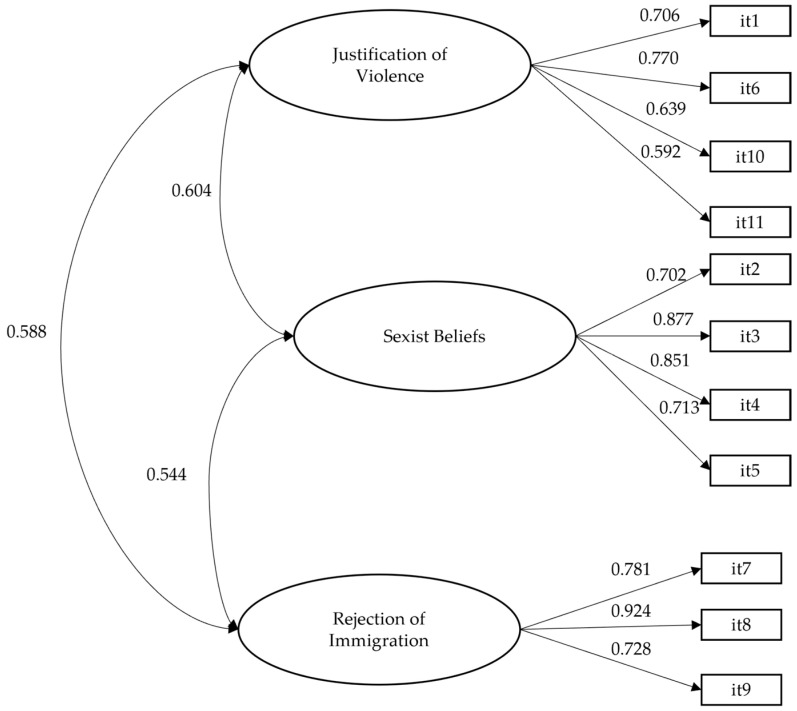
Three-factor model of the confirmatory factor solution. Note: The Figure shows standardised values.

**Table 1 behavsci-15-00756-t001:** Main Characteristics of the Sample.

Variables	Categories	n (%)
Sex	Male	46.5%
	Female	53.5%
Education Level	9th grade	31.7%
	10th grade	23.9%
	11th grade	24.8%
	12th grade	19.6%
Ethnicity	Indigenous (Aymara/Mapuche)	26.6%
	Non-Indigenous	73.4%

**Table 2 behavsci-15-00756-t002:** Descriptive Statistics.

Item	Factors	Mean	Standard Deviation	Skewness	Kurtosis	K-S Test
Item 1	JV	2.47	1.20	0.50	−0.57	0.188 *
Item 2	SB	1.79	1.03	1.38	1.54	0.202 *
Item 3	SB	1.83	1.13	1.37	1.09	0.185 *
Item 4	SB	1.72	1.03	1.53	1.79	0.172 *
Item 5	SB	2.01	1.26	1.02	−0.13	0.293 *
Item 6	JV	2.23	1.20	0.72	−0.36	0.312 *
Item 7	RI	2.32	1.21	0.67	−0.41	0.330 *
Item 8	RI	1.93	1.11	1.19	0.79	0.295 *
Item 9	RI	2.01	1.15	0.96	0.03	0.210 *
Item 10	JV	2.47	1.27	0.40	−0.86	0.262 *
Item 11	JV	2.50	1.25	0.46	−0.70	0.258 *

Note: * *p* < 0.01. SB = sexist beliefs; JV = justification of violence; RI = rejection of immigration. The abbreviations refer to the dimensions assessed by the Authoritarian Attitude Scale and are included to facilitate the interpretation of the results.

**Table 3 behavsci-15-00756-t003:** Measurement invariance.

Variable/Model	WLSMV-χ^2^ (df)	RMSEA	CFI	TLI	SRMR	ΔRMSEA	ΔCFI	ΔTLI	DECISIÓN
Sex									
M0	181.840 (82)	0.056	0.980	0.973	0.034	—	—	—	Accepted
M1	209.795 (93)	0.057	0.977	0.972	0.040	0.001	−0.003	−0.001	Accepted
M2	258.504 (120)	0.055	0.972	0.975	0.038	−0.002	−0.005	0.003	Accepted
M3	314.352 (131)	0.06	0.963	0.969	0.042	0.005	−0.009	−0.006	Accepted
Age *									
M0	173.480 (82)	0.054	0.983	0.978	0.031	—	—	—	Accepted
M1	187.589 (93)	0.051	0.983	0.980	0.037	−0.003	0	0.002	Accepted
M2	226.162 (120)	0.048	0.981	0.982	0.034	−0.003	−0.002	0.002	Accepted
M3	237.195 (131)	0.046	0.981	0.984	0.037	−0.002	0	0.002	Accepted
Ethnicity									
M0	158.137 (82)	0.047	0.987	0.983	0.03	—	—	—	Accepted
M1	184.998 (93)	0.049	0.985	0.982	0.037	0.002	−0.002	−0.001	Accepted
M2	206.185 (120)	0.042	0.986	0.987	0.033	−0.007	0.001	0.005	Accepted
M3	215.133 (131)	0.039	0.986	0.988	0.036	−0.003	0	0.001	Accepted

Note: WLSMV-χ^2^ = weighted least square with mean and variance adjusted; df = degrees of freedom; CFI = comparative fit index; TLI = Tucker–Lewis fit index; RMSEA = root mean square error of approximation; ΔCFI = change in CFI; ΔTLI = change in TLI; M0 = configural; M1 = weak, M2 = strong, M3 = strict. Age * was dichotomized in the first group 14–15 years and second group 16–19 years.

**Table 4 behavsci-15-00756-t004:** Reliability.

Factor	McDonald’s ω (95% CI)	Cronbach’s α (95% CI)	Greatest Lower Bound (95% CI)
Justification of violence	0.719 (CI = 0.686–0.751)	0.719 (CI = 0.685–0.750)	0.756 (CI = 0.718–0.791)
Sexist beliefs	0.798 (CI = 0.775–0.821)	0.795 (CI = 0.771–0.818)	0.809 (CI = 0.780–0.843)
Rejection of immigration	0.801 (CI = 0.778–0.825)	0.796 (CI = 0.769–0.820)	0.801 (CI = 0.767–0.832)

Note: The values in parentheses () are the confidence intervals of the reliability estimations.

## Data Availability

The dataset for the study is available from the corresponding author upon reasonable request due to ethical restrictions.

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
