# Peer review of "Psychometric Properties of the Authoritarian Attitude Scale in a Sample of Chilean Adolescent Students"

_behavsci, 2025, doi:10.3390/bs15060756_

Round 1

Reviewer 1 Report

Comments and Suggestions for Authors

This manuscript presents a scale adaptation study that evaluated the psychometric properties of the Authoritarian Attitude Scale (AAS) in Chilean adolescents. A sample of secondary school students was recruited from various schools to examine the reliability and validity of a brief version of AAS. Factor analysis results and reliability and convergent validity evidence supported the psychometric quality of the three-factor AAS. 

Authoritarianism is a topic of importance in social psychology and may have critical effects on the socioemotional development of adolescent students. The adaptation and validation of the relevant measures have theoretical and practical implications and are potentially interesting to the readers of Behavioral Sciences. Strengths of this paper include a representative sample recruitment and the rigorous examination of factor structure, measurement invariance, and reliability evidence. In addition, the manuscript is generally well-written. I have a few suggestions that may enhance the contribution of this work.

First, within the construct of the authoritarian attitude, this manuscript focuses on three factors--sexism, justification of violence, and rejection of immigration. Why focus on these three factors? What other factors are considered under the construct? The literature lacks the introduction of the overall theoretical framework and the justification for measuring these factors. 

Second, the study aimed to "design and validate a brief scale" (line 148, p.4) based on the AAS developed by Carrión-María et al., 2012. Since the original document was in Spanish, it would be very informative for the readers to provide an overview of the original AAS. Especially the theoretical framework and how it was developed. Section 2.3 mentions that the AAS consists of 11 items and three factors (p.5), which is already a brief scale. What did the current study do to further design this brief scale?

Third, and probably most importantly, the paper should clarify what it contributes to the literature. This study used the same population of secondary school students and the same items as in Carrión-María et al., 2012. Why is it necessary to conduct a validation in a standalone study? What does it contribute the the understanding of the authoritarian attitude? Even though the authors did not intend to conduct a cross-cultural study, it would be helpful to provide some cultural background on the differences between Spain and Chile, or justify the importance of the three factors (sexism, justification of violence, and rejection of immigration) for Chilean adolescents.

Fourth, the evidence for convergent validity was relatively simple.  The only criterion refers to "engagement in criminal behaviors and the use of weapons" (line 196, p.5). Out of all the related constructs discussed in the introduction, it is not obvious how the selected criterion was related to the authoritarian attitude. I also doubt whether this is an appropriate variable for secondary school students in public and government-subsidized schools. The insufficient validity evidence could be discussed as one of the limitations of the current study.

Finally, I have a few specific comments, mostly related to writing.

1. The abstract mentions that participants were recruited from 11 schools, but the sampling details are not provided in section 2.1.
2. Section 2.4 only mentions that CFA was conducted using 7.11. How about measurement invariance checks and reliability estimation? Please also report the software used for the other analyses.
3. In Table 2 (p.6), I would suggest replacing "g1" and "g2" with skewness and kurtosis to reduce the cognitive load for readers. Also, no need to provide the full terms for M and SD.
4. Section 3.4 reports only the correlations between violent behavior and AAS subscales. The means and standard deviations for these four scale scores should also be reported.
5. I don't think the cross-sectional design should be considered a study limitation. This study aims to examine the psychometric properties of a scale. The aim is not to understand the development of authoritarian attitude or investigate its antecedents and consequences. I cannot see the values of longitudinal approaches.

Author Response

For research article

Response to Reviewer 1 Comments

1. Summary

Thank you very much for taking the time to review this manuscript. Please find the detailed responses below and the corresponding revisions.

2. Questions for General Evaluation

Reviewer’s Evaluation

Response and Revisions

Is the content succinctly described and contextualized with respect to previous and present theoretical background and empirical research (if applicable) on the topic?

Must be improved

Thank you for your observations. At this stage, specific improvements have been made following the suggestions provided.

Are the research design, questions, hypotheses and methods clearly stated?

Yes

Thank you for your observations. At this stage, improvements have been made following the suggestions provided.

Are the arguments and discussion of findings coherent, balanced and compelling?

Can be improved

Thank you for your observations. At this stage, specific improvements have been made following the suggestions provided.

For empirical research, are the results clearly presented?

Can be improved

Thank you for your observations. At this stage, specific improvements have been made following the suggestions provided.

Is the article adequately referenced?

Can be improved

Thank you for your observations. At this stage, specific improvements have been made following the suggestions provided.

Are the conclusions thoroughly supported by the results presented in the article or referenced in secondary literature?

Can be improved

Thank you for your observations. At this stage, specific improvements have been made following the suggestions provided.

3. Point-by-point response to Comments and Suggestions for Authors

Reviewer 1.

Comments 1: First, within the construct of the authoritarian attitude, this manuscript focuses on three factors--sexism, justification of violence, and rejection of immigration. Why focus on these three factors? What other factors are considered under the construct? The literature lacks the introduction of the overall theoretical framework and the justification for measuring these factors.

Response 1: We appreciate your valuable observation regarding the need to theoretically justify the choice of the three factors analysed. We have strengthened this dimension in the introduction of the manuscript (lines 71 and 81), integrating a more explicit rationale on why the focus is on sexist beliefs, justification of violence, and rejection of immigration as central dimensions of authoritarian attitudes in adolescence.

Likewise, the justification for measuring these factors has been added in lines (82 to 86).

Comments 2: Second, the study aimed to "design and validate a brief scale" (line 148, p.4) based on the AAS developed by Carrión-María et al., 2012. Since the original document was in Spanish, it would be very informative for the readers to provide an overview of the original AAS. Especially the theoretical framework and how it was developed. Section 2.3 mentions that the AAS consists of 11 items and three factors (p.5), which is already a brief scale. What did the current study do to further design this brief scale?

Response 2: We appreciate your review; this fragment has been removed from the text.

The general version of the original scale is included in the theoretical framework, where it is explicitly stated that the joint incorporation of these factors corresponds to a comprehensive understanding of authoritarianism as a socially configured attitude, as proposed by the original author, Carrión-María et al. (2012). Lines 82–86. Regarding the construction and reliability values of the original scale, these are presented in lines 246–252.

Comments 3: Third, and probably most importantly, the paper should clarify what it contributes to the literature. This study used the same population of secondary school students and the same items as in Carrión-María et al., 2012. Why is it necessary to conduct a validation in a standalone study? What does it contribute the the understanding of the authoritarian attitude? Even though the authors did not intend to conduct a cross-cultural study, it would be helpful to provide some cultural background on the differences between Spain and Chile, or justify the importance of the three factors (sexism, justification of violence, and rejection of immigration) for Chilean adolescents.

Response 3: We sincerely appreciate your observation, which allows us to strengthen the conceptual foundation of the study. Regarding the need to revalidate an instrument previously used in another population (in this case, Spanish adolescents), we consider it essential to point out that validity evidence is not an immutable property of the test itself, but of the interpretations made of the scores obtained in specific contexts. According to the Standards for Educational and Psychological Testing (AERA, APA & NCME, 2014), “validation is not a single event, but a continuous process of accumulating evidence to support specific interpretations of scores for intended uses and particular populations” (p. 11). Therefore, even when using the same items, it is methodologically necessary to accumulate new empirical evidence to support their use in culturally different samples.

This study contributes to the literature by evaluating the psychometric properties of an authoritarian attitude scale in Chilean adolescents, a population underrepresented in empirical studies of this type. Although the same items from Carrión-María et al.'s (2012) study were used, our research provides validity evidence based on the internal structure and score reliability in a different sociocultural context, which is essential for advancing the understanding of authoritarian attitudes from a situated perspective.

We agree on the importance of incorporating a contextual analysis. Although the purpose of the study was not to make an explicit cross-cultural comparison between Spain and Chile, we have incorporated a section that offers an interpretative framework of the Chilean context. In this country, social conservatism, the persistence of gender inequalities, the growing migratory presence and its associated tensions, along with a school system marked by socioeconomic segmentation, configure a relevant context for analysing how authoritarian attitudes manifest in adolescence. These conditions reinforce the relevance of specifically examining the factors of sexism, justification of violence, and rejection of immigration among Chilean youth, providing empirical evidence that can enrich future comparisons with other national realities.

Lines incorporated in the manuscript Line 193 - 200

Comments 4: Fourth, the evidence for convergent validity was relatively simple.  The only criterion refers to "engagement in criminal behaviors and the use of weapons" (line 196, p.5). Out of all the related constructs discussed in the introduction, it is not obvious how the selected criterion was related to the authoritarian attitude. I also doubt whether this is an appropriate variable for secondary school students in public and government-subsidized schools. The insufficient validity evidence could be discussed as one of the limitations of the current study.

Response 4: We sincerely appreciate your observation, which we value as an opportunity to clarify our methodological choice. The use of the criterion of “participation in criminal behaviours and use of weapons” is based on prior empirical evidence that has consistently linked authoritarian attitudes with a greater propensity toward justifying violence and punitive social control, even in school contexts (Altemeyer, 1996; Bobbio & Sarracino, 2011). From this perspective, this criterion is not intended to measure criminal behaviours, but rather to serve as an extreme behavioural indicator that allows for exploring convergence with dimensions such as justification of violence, one of the subscales evaluated.

Local research has shown the presence of attitudes and narratives associated with the instrumental use of force or intimidation, which may be linked to authoritarian school climates or perceptions of threat (Toro et al., 2019).

Nonetheless, we have incorporated an additional brief clarifying comment in both the theoretical framework and the discussion.

(Lines 201-212; 487-491)

Comments 5: The abstract mentions that participants were recruited from 11 schools, but the sampling details are not provided in section 2.1.

Response 5: We sincerely appreciate your observation. We have incorporated a more detailed description of the sampling procedure used in the study in section 2.1 (Participants). Specifically, we have added that the participants were selected through non-probabilistic convenience sampling. Additionally, it has been specified that 11 secondary schools in the Region of La Araucanía were contacted, which expressed their willingness to participate, and that the questionnaires were administered online during school hours, with prior institutional authorisation. This information aims to clearly address your suggestion and strengthen the methodological transparency of the study. (Lines 227 to 229)

Comments 6: Section 2.4 only mentions that CFA was conducted using 7.11. How about measurement invariance checks and reliability estimation? Please also report the software used for the other analyses.

Response 6: Thank you for your observation. We agree that it is important to specify the software used for the additional analyses. We have clarified that the software used for the descriptive and univariate normality analysis was SPSS, the invariance models were run using MPLUS 8.11, and the reliability estimates were computed using JASP 0.19. (Highlighted in yellow in the data analysis section) (Lines 289; 301-302 and 308-309)

Comments 7: In Table 2 (p.6), I would suggest replacing "g1" and "g2" with skewness and kurtosis to reduce the cognitive load for readers. Also, no need to provide the full terms for M and SD.

Response 7: Thank you for your comment. We have incorporated your observations into Table 2, replacing the technical abbreviations with more accessible terms and simplifying the presentation of the descriptive statistics. (Lines 321-322)

Comments 8: Section 3.4 reports only the correlations between violent behavior and AAS subscales. The means and standard deviations for these four scale scores should also be reported.

Response 8: Thank you for your observation. We have supplemented section 3.4 by incorporating the means and standard deviations corresponding to the four scale scores. (Lines 355-358)

Comments 9: I don't think the cross-sectional design should be considered a study limitation. This study aims to examine the psychometric properties of a scale. The aim is not to understand the development of authoritarian attitude or investigate its antecedents and consequences. I cannot see the values of longitudinal approaches.

Response 9: We appreciate your comment. In response to your suggestion, we have removed the reference to longitudinal designs in the discussion section.

4. Response to Comments on the Quality of English Language

Point 1:

Response 1 The English is fine and does not require any improvement.

5. Additional clarifications

We sincerely appreciate your valuable observations. We have made the corresponding changes according to the reviewer’s suggestions.

Reviewer 2 Report

Comments and Suggestions for Authors

I will offer my mixed reactions to this manuscript. The rationale behind the study is well presented and the writing style follow a logical and straightforward trajectory. Literature references are current and comprehensive. Regarding the methodology, the sample is well described and is sufficiently large enough for the conducted analyses. Results are clearly presented and the Discussion, for the most part, is supported by the data.

The reservations I do have relate the methodology of the study. Regarding the main instrument of the study, the Authoritarian Attitude Scale, the notion that 11 items can adequately assess three unique constructs can be problematic. With few items, each factor may not fully represent the breadth or complexity of its intended construct (e.g., "sexist beliefs" might be multifaceted but underrepresented by only 3 items). Having exactly 3 items per factor is the minimum — which allows for model identification but not much room for testing alternative structures or allowing for correlated errors.

Similarly, regarding the reported reliability it the original scale, while "adequate" is fine, differentiating between “acceptable” (0.70–0.79) and “good” (0.80–0.89) reliability would give readers a clearer picture.

Regarding the calculated CFA, Several key fit indices are reported, which gives a well-rounded view of model adequacy:

  • WLSMV-χ² (weighted least squares mean and variance adjusted): Chi-square is significant, but this is expected with larger samples.
  • RMSEA = 0.048 (90% CI = 0.038–0.059): Below 0.05 suggests a close fit.
  • CFI = 0.986; TLI = 0.982: Both > 0.95 indicate an excellent fit.

However, there’s no discussion of why the three factors (justification of violence, sexist beliefs, rejection of immigration) were hypothesized. Was this grounded in theory, prior research, or an exploratory factor analysis (EFA)? Additionally, the summary doesn't address whether any modification indices were used or if items had significant cross-loadings or correlated residuals—important in refining model fit and interpreting factor purity. Further, there’s no mention of whether alternative models (e.g., one-factor, bifactor, or hierarchical models) were tested or ruled out. This limits our confidence in the three-factor solution being the best representation of the data.

Next, in the factorial invariance analyses, age was dichotomized into 14–15 vs. 16–19, but there’s no justification for this cutoff. Why these specific age brackets? Were they based on developmental stages, previous studies, or data distribution? Additionally, In the sex model, the ΔCFI between M2 and M3 is -0.009, which is just at the threshold (commonly, ΔCFI ≤ 0.01 is acceptable). Though this is still acceptable, it could warrant a brief note or discussion. Finally, while statistical invariance is demonstrated, the practical implications are not addressed.

Regarding the reported reliability estimates, all reported metrics are internal consistency estimates, which are useful but limited. There’s no information about test-retest reliability, which would be essential for evaluating stability over time, especially in adolescent populations.

Author Response

 3. Point-by-point response to Comments and Suggestions for Authors

Reviewer 1.

Comments 1: The reservations I do have relate the methodology of the study. Regarding the main instrument of the study, the Authoritarian Attitude Scale, the notion that 11 items can adequately assess three unique constructs can be problematic. With few items, each factor may not fully represent the breadth or complexity of its intended construct (e.g., "sexist beliefs" might be multifaceted but underrepresented by only 3 items). Having exactly 3 items per factor is the minimum — which allows for model identification but not much room for testing alternative structures or allowing for correlated errors.

Response 1: We appreciate your observation regarding the limited number of items per factor in the Authoritarian Attitudes Scale. We agree that representing complex constructs with few items can restrict their conceptual coverage and limit the model's flexibility. However, we would like to point out that this abbreviated version was designed for applied studies, simultaneously measuring multiple scales, with an emphasis on parsimony and measurement efficiency. Each set of items per factor was carefully selected based on theoretical and psychometric criteria, aiming to synthetically represent the core aspects of each dimension (for example, in the case of sexist beliefs, items reflecting explicit gender attitudes were included).

In addition, the measurement model showed satisfactory fit to the data, providing empirical evidence in favour of its structure. Nevertheless, we acknowledge as a limitation the inability to explore alternative structures or allow residual correlations between items. This limitation has been incorporated into the discussion of the manuscript, and we suggest that future research could expand the number of items per dimension to more thoroughly assess construct validity. (Lines 459-464)

Comments 2: Similarly, regarding the reported reliability it the original scale, while "adequate" is fine, differentiating between “acceptable” (0.70–0.79) and “good” (0.80–0.89) reliability would give readers a clearer picture.

Response 2: We sincerely appreciate your valuable observation regarding the need to specify the reliability levels of the factors more precisely. In response to your comment, we have incorporated a more detailed description of the reliability coefficients, specifying whether they correspond to acceptable or good reliability according to the ranges commonly accepted in literature. This information has been included between lines 246 and 252 of the manuscript.

Comments 3: Regarding the calculated CFA, Several key fit indices are reported, which gives a well-rounded view of model adequacy:

WLSMV-χ² (weighted least squares mean and variance adjusted): Chi-square is significant, but this is expected with larger samples.

RMSEA = 0.048 (90% CI = 0.038–0.059): Below 0.05 suggests a close fit.

CFI = 0.986; TLI = 0.982: Both > 0.95 indicate an excellent fit.

However, there’s no discussion of why the three factors (justification of violence, sexist beliefs, rejection of immigration) were hypothesized. Was this grounded in theory, prior research, or an exploratory factor analysis (EFA)? Additionally, the summary doesn't address whether any modification indices were used or if items had significant cross-loadings or correlated residuals—important in refining model fit and interpreting factor purity. Further, there’s no mention of whether alternative models (e.g., one-factor, bifactor, or hierarchical models) were tested or ruled out. This limits our confidence in the three-factor solution being the best representation of the data.

Response 3: The approach we followed in this study was confirmatory. The central objective was to test the theoretical three-factor structure proposed by Carrión-María et al. (2012), which includes the factors of justification of violence, sexist beliefs, and rejection of immigration. This configuration is supported by a solid theoretical and empirical foundation, having been previously validated in an adolescent population. Since the main purpose of the research was to assess the adequacy of this structure in the Chilean context, an exploratory factor analysis (EFA) was not conducted, nor were alternative models (unifactorial, bifactorial, or hierarchical) proposed, as this was a targeted structural validation.

Nevertheless, in agreement with your suggestion, we acknowledge that it would be appropriate in future research to comparatively explore alternative factorial structures and evaluate modification indices as well as potential residual correlations or cross-loadings. These analyses would allow for a more thorough refinement of the model and a better understanding of the internal organisation of the construct.

Additionally, considering the relevance of your comment, we have decided to modify the manuscript title to more precisely reflect the study's confirmatory approach.
(A paragraph has been included in the discussion as suggestions for future research) lines 469-478)-

Comments 4: Next, in the factorial invariance analyses, age was dichotomized into 14–15 vs. 16–19, but there’s no justification for this cutoff. Why these specific age brackets? Were they based on developmental stages, previous studies, or data distribution? Additionally, In the sex model, the ΔCFI between M2 and M3 is -0.009, which is just at the threshold (commonly, ΔCFI ≤ 0.01 is acceptable). Though this is still acceptable, it could warrant a brief note or discussion. Finally, while statistical invariance is demonstrated, the practical implications are not addressed.

Response 4: We appreciate your valuable comment. The cutoff point for the age variable was defined based on the theoretical distinction between early adolescence (14–15 years) and late adolescence (16–19 years), a classification widely supported in developmental literature and aligned with the characteristics of both the instrument and the target population. Additionally, we recognize that the WLSMV estimator imposes specific requirements regarding sample size, particularly for factorial invariance testing. Therefore, efforts were made to ensure sufficiently large and balanced subsamples to guarantee the stability and validity of the obtained estimates.

Regarding the invariance analyses by sex, age, and ethnic identity, the changes in CFI (ΔCFI) observed between increasingly constrained models remained within the commonly accepted threshold (ΔCFI ≤ 0.01). Specifically, for the sex-based model, the ΔCFI between M2 and M3 was −0.009. Although this value is close to the threshold, it is important to note that −0.009 represents a smaller difference in model fit than the limit of 0.01, and thus clearly satisfies the criterion. As suggested by Cheung and Rensvold (2002) and Chen (2007), changes in CFI less than or equal to 0.01 in absolute value are considered evidence of invariance. Therefore, the observed result supports metric invariance across sex.

Finally, the practical implications of the demonstrated statistical invariance have been incorporated into the conclusion section, emphasizing that the scale’s scores can be meaningfully compared across sex, age, and ethnic groups.

Lines 523–529.

Comments 5: Regarding the reported reliability estimates, all reported metrics are internal consistency estimates, which are useful but limited. There’s no information about test-retest reliability, which would be essential for evaluating stability over time, especially in adolescent populations.

Response 5: We appreciate your observation. Indeed, since the present study is based on a cross-sectional design, it was not possible to estimate test-retest reliability, as this requires repeated measurements over time. Nevertheless, we acknowledge the importance of assessing the temporal stability of the instrument, particularly in adolescent populations, and we suggest that future follow-up studies consider including this estimate as part of the comprehensive psychometric validation of the scale. (Lines 474-479).

4. Response to Comments on the Quality of English Language

Point 1:

Response 1 The English is fine and does not require any improvement.

5. Additional clarifications

Reviewer 3 Report

Comments and Suggestions for Authors

Thank you for the opportunity to review “Exploring the Psychometric Properties of the Authoritarian Attitude Scale in a Sample of Chilean Adolescent Students”, which was a psychometric validation study of the aforementioned scale. After a careful review of the manuscript, I have only one substantive comment for the authors:

In the materials and methods section, the authors report three demographic characteristics: Sex, Education level (e.g., age), and ethnicity. They then go on to report the results of measurement invariance tests for both sex and age. Why not also include ethnicity? Since one of their dimensions is rejection of immigration, it seems plausible to me that responses might vary by whether or not respondents are indigenous or non-indigenous. The sample sizes are a little lopsided for these two groups (26.6% vs. 73.4%), but I do not see anything in particular that precludes the authors from examining measurement invariance for this characteristic as well. I encourage them to do so or at least provide a rationale for not doing so.

Beyond that, I have no other comments. I thank the authors for their important research.

Author Response

3. Point-by-point response to Comments and Suggestions for Authors

Reviewer 1.

Comments 1: In the materials and methods section, the authors report three demographic characteristics: Sex, Education level (e.g., age), and ethnicity. They then go on to report the results of measurement invariance tests for both sex and age. Why not also include ethnicity? Since one of their dimensions is rejection of immigration, it seems plausible to me that responses might vary by whether or not respondents are indigenous or non-indigenous. The sample sizes are a little lopsided for these two groups (26.6% vs. 73.4%), but I do not see anything in particular that precludes the authors from examining measurement invariance for this characteristic as well. I encourage them to do so or at least provide a rationale for not doing so.

Response 1: We appreciate the observations, which have significantly contributed to the improvement of our manuscript. It should be noted that they were incorporated into the following sections: theoretical framework (lines 59-66); in the data analysis section, the requested information regarding ethnicity was added (lines 345-346); and the results and discussion section (line 411).

The implications of ethnic invariance are included in the conclusion section (lines 523-529).

4. Response to Comments on the Quality of English Language

Point 1:

Response 1 The English is fine and does not require any improvement.

5. Additional clarifications

We sincerely appreciate your valuable observations. We have made the corresponding changes according to the reviewer’s suggestions.

Reviewer 4 Report

Comments and Suggestions for Authors

Author Response

3. Point-by-point response to Comments and Suggestions for Authors

Reviewer 1.

Comments 1: The phrase, “which are socially unacceptable,” in line 76, is not needed. Readers with any sense of morality know that the behaviors described in the previous sentence are unacceptable and the sentence has a stronger impact if it simply states, “These behaviors can cause both physical and

psychological harm…”

Response 1: We greatly appreciate your suggestion. We fully agree that the sentence becomes more forceful and precise by removing the explicit evaluation of the behaviours as “socially unacceptable.” In response, we have revised the wording to express the impact of these behaviours more directly.
Included between lines 101-105.

Comments 2: Can the authors clarify what they mean by “in secondary education,” in line 124? Does this phrase refer to secondary education as a cultural institution or to adolescents in the secondary education years of schooling?

Response 2: We sincerely appreciate your observation. When referring to “secondary education,” we specifically allude to adolescents enrolled in the levels corresponding to secondary education. For greater clarity, we have rephrased the sentence in the manuscript as follows:

“The scientific literature has shown that the combination of sexist beliefs, justification of violence, and rejection of immigration in adolescents enrolled in secondary education has multiple negative consequences…”

This modification aims to eliminate ambiguities and precisely specify the group to which we are referring.

Lines 149-150.

Comments 3: It seems that a dimension of authoritarianism should pertain to bias against racial minorities, as a counterpart to the dimension related to sexism. Is there a reason why the construct of racial bias is not included among the dimensions measured in the Authoritarian Attitude Scale or addressed in the evaluation of it conducted by the authors of this manuscript?

Response 3: We appreciate your observation. In relation to your comment, it is important to note that the Authoritarian Attitudes Scale used in this study is based on the original model developed by Carrión-María et al. (2012), which conceptualises authoritarianism in adolescents through three specific dimensions: sexist beliefs, justification of violence, and rejection of migration. While we acknowledge that bias toward racial minorities is a relevant component in classical research on authoritarianism (for example, Adorno et al., 1950), the original authors chose to address forms of social exclusion particularly relevant in contemporary Spanish-speaking contexts, prioritising migration as a concrete expression of rejection of the “other.”

Regarding our work, since our objective was to validate the factorial structure proposed in the original scale, we did not incorporate additional dimensions or modify the theoretical model. Nevertheless, we agree that the future inclusion of a specific component addressing racial bias could enrich the understanding of authoritarianism in sociocultural contexts characterised by greater ethnic diversity, and we therefore consider your suggestion highly relevant for future research. Lines 473-475.

Comments 4: In Table 2,

a. the leftmost column should refer to Item 1, Item 2, Item 3, etc., not Items (unless I have

misinterpreted).

b. the statement associated with each item or, at least, the construct that each addresses should

appear. This information would not only provide much-appreciated context within Table 2 but

would also increase reader understanding of the path diagram (factor model) that appears in

Section 3.2.

Response 4: We appreciate your observation and have proceeded to correct the noted error. The statements corresponding to each item are available in the supplementary material.

Comments 5: Despite having very low p values, the three correlation coefficients presented in Section 3.4 indicate only weak to moderate relationships between the factors evaluated. The authors should explain why these weak to moderate relationships still reflect strong patterns in the variability of scores for the factors involved in each analysis.

Response 5: We appreciate your observation and agree that the correlation coefficients observed indicate relationships of low to moderate magnitude between the variables studied. However, in psychological and social research, especially when analysing complex constructs such as attitudes or beliefs, it is common to observe modest correlations that are nonetheless significant and conceptually meaningful (Gignac & Szodorai, 2016). (Lines 358-365)

In this context, the reported correlations reflect consistent and theoretically expected patterns: participants who score higher on violent behaviours also tend to justify violence more, hold stronger sexist beliefs, and show greater rejection of immigration. Although the absolute r values are not high, these results provide empirical evidence supporting the coexistence of authoritarian attitudes and violent behaviours, aligning with previous findings in the literature (Altemeyer, 1996; Sibley & Duckitt, 2008).

Comments 6: It doesn’t come as much of a surprise that the Antisocial and Delinquent Behavior scale’s violent behavior factor correlates most strongly with the Authoritarian Attitude Scale’s justification of violence dimension. The violent behavior factor’s strong associations with sexist beliefs and rejection of immigration, however, is not so predictable. These relationships raise important questions about violent manifestations of bias. The results of this study suggest that the three dimensions measured by the Authoritarian Attitude Scale do not simply coexist in those who have authoritarian personalities.

If they simply coexisted, then each would operate independently. But, the relationships that all three dimensions have with Antisocial and Delinquent Behavior scale item suggest that they do not operate independently. Rather, those with sexist beliefs can be expected to accept violent behavior more easily that those without such beliefs do. Similarly, those with low tolerance for immigrants can be expected to accept violent behavior more easily than those with high tolerance for immigrants do. The evidence of these relationships, I believe, has just as much value in studies of human behavior as the

confirmation of the Authoritarian Attitudes Scale’s multi-cultural applicability does. Therefore, I encourage the authors to add comments in the manuscript (most likely in the Discussion or Conclusion section) that stress the interplay, between the three factors measured by the Authoritarian Attitude Scale.

Response 6: We appreciate your comments and have incorporated the observations provided into the manuscript, highlighting the interrelationship between the three factors measured by the Authoritarian Attitudes Scale in the discussion, as noted in lines 388-394.

4. Response to Comments on the Quality of English Language

Point 1:

Response 1 The English is fine and does not require any improvement.

5. Additional clarifications

We sincerely appreciate your valuable observations. We have made the corresponding changes according to the reviewer’s suggestions.

Round 2

Reviewer 1 Report

Comments and Suggestions for Authors

The revised manuscript has made substantial improvements. Responses to my concerns are also satisfactory. I now feel comfortable recommending this manuscript for publication pending a minor revision suggestion regarding the terminology. I would recommend replacing "convergent validity" with "criterion validity". On one hand, violent behavior is conceptually different from the authoritarian factors. On the other hand, the correlation is of a small effect size. For convergent evidence, a larger relationship would be expected.

Author Response

3. Point-by-point response to Comments and Suggestions for Authors

Reviewer 1.

Comments 1: The revised manuscript has made substantial improvements. Responses to my concerns are also satisfactory. I now feel comfortable recommending this manuscript for publication pending a minor revision suggestion regarding the terminology. I would recommend replacing "convergent validity" with "criterion validity". On one hand, violent behavior is conceptually different from the authoritarian factors. On the other hand, the correlation is of a small effect size. For convergent evidence, a larger relationship would be expected.

Response 1: We sincerely appreciate your positive feedback and the time you devoted to reviewing our manuscript. We are pleased to know that the revisions made have been satisfactory. In response to your valuable suggestion regarding terminology, we have replaced the term “convergent validity” with “criterion validity,” which we believe adds greater conceptual precision to the text. This modification is reflected in lines 28, 351, 420, 445, and 489.

Regarding the conceptual framing of violent behavior, this point has been incorporated between lines 209 and 211, where it is stated that, on one hand, violent behavior is conceptually distinct from authoritarian factors, but its connection is acknowledged based on Altemeyer’s (1996) framework.

Indeed, we agree that violent behavior and authoritarian attitudes are conceptually distinct constructs, although the literature has documented a functional relationship between them, particularly in relation to the justification of violence and social control.

We also appreciate your observation regarding the effect sizes of the reported correlations. Although the observed coefficients fall within the small to moderate range according to Cohen’s (1988) criteria, these findings are consistent with research in the field of social psychology, where constructs tend to be multifactorial and associations typically show modest effect sizes (Gignac & Szodorai, 2016). In this sense, the statistical significance achieved, along with the theoretical coherence among the variables analyzed, supports the idea that individuals who engage in more violent behavior also tend to exhibit attitudes that legitimize violence and authoritarian stances. Nevertheless, in light of your valuable suggestion, we have adjusted the manuscript’s wording to temper the interpretation of this relationship, acknowledging its limitations and contextualizing it within the existing literature. This revision has been incorporated between lines 365 and 368.

Reviewer 2 Report

Comments and Suggestions for Authors

I have read the revised manuscript and can now support publication in Behavioral Sciences. I very much appreciate the author’s attention to detail in considering my recommendations and believe it has produced a stronger, overall manuscript. While most of my comments focused on the methodology section of the manuscript, this revision also incorporates a number of important additions in both the Introduction and the Discussion sections.  The overall tone of the manuscript, acknowledging some of the psychometric limitations of the instrument, has also been sufficiently strengthened. 

I would make several final additional comments:

  1. The final sentence of the abstract “The results indicate that the scale is valid and reliable for assessing authoritarian attitudes in Chilean high school students.”, should be tempered a bit as not all types of reliability and validity were examined.
  2. On lines 185 and 186, the two added citations are not in alphabetical order.
  3. Same comment on lines 208 and lines 211-212.

Overall, a very solid revision.

Author Response

3. Point-by-point response to Comments and Suggestions for Authors

Reviewer 1.

Comments 1: I have read the revised manuscript and can now support publication in Behavioral Sciences. I very much appreciate the author’s attention to detail in considering my recommendations and believe it has produced a stronger, overall manuscript. While most of my comments focused on the methodology section of the manuscript, this revision also incorporates a number of important additions in both the Introduction and the Discussion sections.  The overall tone of the manuscript, acknowledging some of the psychometric limitations of the instrument, has also been sufficiently strengthened

I would make several final additional comments:

The final sentence of the abstract “The results indicate that the scale is valid and reliable for assessing authoritarian attitudes in Chilean high school students.”, should be tempered a bit as not all types of reliability and validity were examined.

Response 1: We greatly appreciate your comment. As you suggested, we have revised the final statement in the abstract to more accurately reflect the scope of the evidence presented. This modification has been incorporated in lines 31 and 33 of the manuscript.

Comments 2: On lines 185 and 186, the two added citations are not in alphabetical order.

Response 2: Thank you for the observation. The alphabetical order of the citations in lines 185 and 186 has been corrected in the revised version of the manuscript.

Comments 3: 3.           Same comment on lines 208 and lines 211-212.

Overall, a very solid revision.

Response 3: Thank you for the observation. The alphabetical order of the citations in lines 185 and 186 has been corrected in the revised version of the manuscript. Likewise, the alphabetical order of the citations in lines 209 and in lines 212 and 213 has also been adjusted.

Reviewer 3 Report

Comments and Suggestions for Authors

Thank you for the opportunity to re-review the manuscript now titled “Exploring the Psychometric Properties of the Authoritarian Attitude Scale in a Sample of Chilean Adolescent Students”. The authors addressed my sole comment from my first reading of the paper. I now only have minor edits to offer.

In the discussion section, on page 12, the authors include new content regarding their CFA. This appears to be in response to another reviewer. My personal opinion is that the authors do not need to suggest the use of modification indices and examining cross-loadings and correlated residuals. Their CFA had good model fit from the beginning. The three previous elements are only for models with poor initial fit. While I agree that they can note the limitation of not having alternative models to test, the rest of the information is not needed in my view. Other reviewers may feel differently.

Either way though, the authors accidentally included what appears to be a response to a reviewer in their actual manuscript: “Likewise, we consider your suggestion highly relevant for future research, agreeing that the inclusion of a component addressing racial bias could enrich the understanding of authoritarianism”. This should absolutely be removed from the manuscript.

Beyond these points, I have no additional comments. I applaud the authors on their important work.

Author Response

3. Point-by-point response to Comments and Suggestions for Authors

Reviewer 1.

Comments 1: Thank you for the opportunity to re-review the manuscript now titled “Exploring the Psychometric Properties of the Authoritarian Attitude Scale in a Sample of Chilean Adolescent Students”. The authors addressed my sole comment from my first reading of the paper. I now only have minor edits to offer.

In the discussion section, on page 12, the authors include new content regarding their CFA. This appears to be in response to another reviewer. My personal opinion is that the authors do not need to suggest the use of modification indices and examining cross-loadings and correlated residuals. Their CFA had good model fit from the beginning. The three previous elements are only for models with poor initial fit. While I agree that they can note the limitation of not having alternative models to test, the rest of the information is not needed in my view. Other reviewers may feel differently.

Response 1: We sincerely appreciate your valuable observation. In accordance with your suggestion, we have removed the content referring to modification indices, cross-loadings, and correlated residuals from page 12 of the manuscript, as you indicated. We agree that these procedures are unnecessary given the good initial fit of our model, and their inclusion could have caused confusion regarding the methodological approach adopted. We believe your comment was highly relevant and contributed to improving the clarity and rigor of the manuscript. Once again, we thank you for your thorough review and for your contributions to strengthening this work.

Comments 2: Either way though, the authors accidentally included what appears to be a response to a reviewer in their actual manuscript: “Likewise, we consider your suggestion highly relevant for future research, agreeing that the inclusion of a component addressing racial bias could enrich the understanding of authoritarianism”. This should absolutely be removed from the manuscript.

Beyond these points, I have no additional comments. I applaud the authors on their important work.

Response 2: Indeed, it was a response to the reviewer that was mistakenly included in the manuscript. We have now removed the following sentence from the text: “Likewise, we consider your suggestion highly relevant for future research, agreeing that the inclusion of a component addressing racial bias could enrich the understanding of authoritarianism.” We sincerely appreciate your attentive comment, which has helped to improve the quality of the manuscript.